# Triptolide, a Cancer Cell Proliferation Inhibitor, Causes Zebrafish Muscle Defects by Regulating Notch and STAT3 Signaling Pathways

**DOI:** 10.3390/ijms25094675

**Published:** 2024-04-25

**Authors:** Byongsun Lee, Yongjin Park, Younggwang Lee, Seyoung Kwon, Jaekyung Shim

**Affiliations:** 1Department of Bioresources Engineering, Sejong University, Seoul 05006, Republic of Korea; coolbs@gmail.com (B.L.); yongjinp3@gmail.com (Y.P.); shfdlxj31@naver.com (Y.L.); ksy21333@naver.com (S.K.); 2Institute of Medical Science, Ajou University School of Medicine, Suwon 16499, Republic of Korea

**Keywords:** triptolide, STAT3, Notch1, cancer cell, muscle atrophy, muscle development, zebrafish

## Abstract

Triptolide is a natural compound in herbal remedies with anti-inflammatory and anti-proliferative properties. We studied its effects on critical signaling processes within the cell, including Notch1 and STAT3 signaling. Our research showed that triptolide reduces cancer cell proliferation by decreasing the expression of downstream targets of these signals. The levels of each signal-related protein and mRNA were analyzed using Western blot and qPCR methods. Interestingly, inhibiting one signal with a single inhibitor alone did not significantly reduce cancer cell proliferation. Instead, MTT assays showed that the simultaneous inhibition of Notch1 and STAT3 signaling reduced cell proliferation. The effect of triptolide was similar to a combination treatment with inhibitors for both signals. When we conducted a study on the impact of triptolide on zebrafish larvae, we found that it inhibited muscle development and interfered with muscle cell proliferation, as evidenced by differences in the staining of myosin heavy chain and F-actin proteins in confocal fluorescence microscopy. Additionally, we noticed that inhibiting a single type of signaling did not lead to any significant muscle defects. This implies that triptolide obstructs multiple signals simultaneously, including Notch1 and STAT3, during muscle development. Chemotherapy is commonly used to treat cancer, but it may cause muscle loss due to drug-related adverse reactions or other complex mechanisms. Our study suggests that anticancer agents like triptolide, inhibiting essential signaling pathways including Notch1 and STAT3 signaling, may cause muscle atrophy through anti-proliferative activity.

## 1. Introduction

Over the years, scientists have been studying various ways to develop effective anticancer drugs. In this particular research, our goal was to investigate the molecular mechanisms underlying the antiproliferative activity of triptolide (TP) in cancer cells and determine its potential as a practical anticancer agent. TP is a natural compound extracted from the Chinese herb *Tripterygium wilfordii* Hook F (TWHF). It exhibits various biological activities such as anti-inflammatory, antipyretic, and immunosuppressive effects [1,2,3].

Recent studies have also shown that TP has potent anticancer properties [4,5,6]. Although TP has been found to inhibit the proliferation and induce the apoptosis of various cancer cells, the exact molecular mechanisms are not yet clear.

Studies related to its molecular mechanisms have shown that TP can regulate the STAT3 pathway, which is frequently activated in cancer and is a potential target for cancer therapy [7,8,9,10]. Research has shown that TP can limit STAT3 phosphorylation [11,12,13], thereby reducing STAT3-mediated gene expression and affecting the proliferation and survival of cancer cells [8,14,15,16]. 

Prior to commencing this study, we conducted a screening of 1100 natural and FDA-approved compounds with the aim of discovering substances that exhibit anticancer effects by inhibiting Notch1 signaling. After conducting experiments, we confirmed that TP inhibits Notch1 signaling. Our findings have led us to believe that TP has the potential to exert potent anticancer effects by regulating both Notch and STAT3 signaling pathways. Therefore, these findings suggest that TP has the potential to exert its anticancer effects by regulating both Notch and STAT3 signaling pathways.

The target genes shared by the STAT3 and Notch1 signaling pathways have not been fully defined yet and are currently being researched. However, some studies suggest that the target genes between the two pathways may overlap or converge [17,18]. For instance, the Notch1 pathway can regulate gene expression in cell proliferation and survival, such as *Cyclin D1* and *Bcl-2* [19,20,21,22]. Likewise, the STAT3 pathway has been shown to regulate the expression of similar function genes, such as *c-Myc* and *Survivin* [15,23,24,25]. Both the Notch1 and STAT3 pathways are involved in the regulation of angiogenesis, and both have been shown to regulate the expression of genes like VEGF that are involved in this process [26,27]. Overall, the target genes of the Notch1 and STAT3 pathways are complex and likely interdependent, and further studies are needed to fully understand the relationship between these two signaling pathways.

Our experimental results have shown that TP has the potential to be an effective anticancer agent by inhibiting essential signals that are involved in cancer cell proliferation, such as Notch1 and STAT3 signaling. To ensure the safety of anticancer drugs, it is crucial to evaluate their toxicity. Zebrafish (*Danio rerio)* have been recognized as a valuable drug screening model for toxicity and safety assessment due to their physiological, morphological, and histological similarities to mammals [28,29]. An evaluation of TP in zebrafish revealed a potential side effect of muscle defect caused by an anti-proliferative property in muscle development. This muscle defect seems mainly caused by the inhibition of muscle cell growth, as seen from the anti-proliferative effects of TP at the cellular level.

Skeletal muscle is a highly adaptable tissue that can change its mass, function, and metabolism in response to various internal and external factors, which is crucial for maintaining muscle homeostasis and overall health [30,31]. Muscle homeostasis changes after birth due to muscle regeneration, aging, and disease progression. At this time, the homeostasis of muscle stem cells also changes, which is related to the inhibition of satellite cell proliferation [30]. During muscle formation, satellite stem cells proliferate into muscle cells called myoblasts, which then differentiate into final myotubes. Muscle homeostasis may be disrupted during cancer treatment, ultimately leading to muscle loss and the reduced efficacy of chemotherapy treatment for cancer patients [31].

Muscle loss is caused by an imbalance between protein synthesis and protein breakdown induced by many factors, such as aging, oxidative stress, inflammation and [32]. Pathways that control protein degradation, such as the ubiquitin proteolytic system and autophagy, are directly responsible for muscle atrophy [32,33]. Studies have shown that increased protein breakdown is the main cause of muscle wasting. However, muscle atrophy is a complex process involving simultaneous regulation of anabolic and catabolic pathways [33]. Therefore, research should not only focus on protein degradation but also consider other factors.

Our hypothesis was that suppressing the proliferation of these muscle cells could contribute to muscle loss by activating protein breakdown, which is one of the leading causes of muscle loss in adults. In this regard, using the zebrafish system to test TP’s toxicity is consistent with our experiment’s objective as it can effectively observe the muscle formation process.

Our study suggests that the changes in muscle cells induced through chemotherapy, often considered side effects, occur via a canonical mechanism that inhibits the growth of new muscle cells required for muscle maintenance. This is achieved by inhibiting the Notch1 and STAT3 signaling pathways, which are also involved in cancer cell proliferation. It also indicates that developing TP as an anticancer drug requires combined methods to minimize potential side effects, such as muscle loss.

## 2. Results 

### 2.1. TP Inhibits Both STAT3 and Notch1 Signal

To investigate whether TP inhibits vital signals such as STAT3 and Notch1 signaling, HeLa cells were treated with P6 (a STAT3 signaling-specific inhibitor) and DAPT (a Notch1 signaling-specific inhibitor) to identify downstream targets of these signaling pathways. HeLa cells are originally derived from cervical cancer cells that have been studied extensively. Previous research has shown that Notch signaling plays a role in cancer development in HeLa cells [34,35].

We measured the mRNA and protein levels of the targets for each signal to confirm the experiment’s validity. The results showed that the P6 treatment decreased the mRNA expression levels of *STAT3*, *Bcl2*, and *Survivin*. However, this treatment did not cause any changes in the mRNA expression of *Notch1*, which is a target of Notch1 signaling [36] (Figure 1A). The protein amount of each gene showed a consistent trend, with pSTAT3 and Survivin decreasing, while the amount of NICD, the active form of the Notch1 protein, remained unchanged (Figure 1B).

Treatment with DAPT, a γ-secretase inhibitor, reduced *Notch1* mRNA expression levels along with *Hes1* and *Hes5*, which are downstream targets of Notch1 signaling. However, it did not have any effect on *STAT3*, a downstream target of STAT3 signaling [37] (Figure 1C). These results indicated a similar pattern: NICD and Hes1 exhibited a reduction at the protein level, while the level of pSTAT3 remained unchanged (Figure 1D). According to the results, P6 and DAPT inhibitors appear to be standardized inhibitors that affect each signal.

We conducted experiments on HeLa cells using equal concentrations of TP as each inhibitor. Our objective was to observe if TP could inhibit the signaling of Notch1 and STAT3 (Figure 1E,F). Our findings revealed that TP was successful in reducing the mRNA expression of *Notch1* and *Hes1*, which are the target genes of Notch1 signaling (Figure 1E). It also inhibited STAT3 signaling, suppressing *STAT3* and *Bcl2* (Figure 1E). The protein abundances of NICD and Hes1 in Notch1 signaling were decreased, while the protein abundances of pSTAT3 and Survivin in STAT3 signaling were also reduced (Figure 1F). Based on these results, we conclude that TP has the ability to simultaneously inhibit the signaling of STAT3 and Notch1.

### 2.2. TP Reduces the Viability of Various Cancer Cells

Various cancer cells were treated with TP and each signaling inhibitor to determine whether TP affects cell viability by inhibiting STAT3 and Notch1 signaling. The MTT assay results of HeLa (derived from cervical cancer) cells show that TP significantly reduces the viability of HeLa compared to the case of simultaneous treatment with P6 and DAPT. HeLa cells have been widely used in the early stages of selecting and testing the therapeutic potential of developing anticancer drugs. They are particularly useful in testing the susceptibility of anticancer drugs [35,38]. In this context, HeLa cells were selected to evaluate cell viability during treatment with TP. The experiment showed that survival decreased most when TP was applied and there was a smaller decrease when both inhibitors were used together. However, surprisingly, using a single treatment of P6 or DAPT did not have a significant effect on cell viability. 

In order to verify whether this phenomenon occurs in specific cells or in various cells, including cancer cells, previous studies were analyzed that tested the pharmacological effects of TP [39,40,41]. Based on the findings, cell viability test were carried out on L929 (noncancerous murine skin fibroblasts) [39], U251 (glioma cells) [41], and Jurkat cells (derived from T-Cell Lymphocyte leukemia) [40]. The same trends were observed in both L929 and U251 (Figure 2B,C). CCK-8 analysis on suspended Jurkat cells also provided similar results (Figure 2D). These results suggest that TP may more effectively interfere with cancer cell proliferation by inhibiting a common downstream target of both STAT3 and Notch signals.

### 2.3. TP Reduces c-Myc and Cyclin d1 mRNA and These Proteins in Cell Proliferation 

*Cyclin d1* and *c-Myc* are essential genes for cell cycle and proliferation. We conducted a study to determine whether these genes are common downstream targets of STAT3 and Notch1 signaling pathways and whether TP can inhibit them. We treated HeLa cells with STAT3 and Notch1 signaling pathway inhibitors and observed a significant decrease in *Cyclin d1* and *c-Myc* mRNA levels (Figure 3A,B). Interestingly, the expression of these genes was slightly reduced with the combined application of P6 and DAPT compared to treatment with each inhibitor individually, but not to the same extent as the individual inhibitors (Figure 3A,B). On the other hand, it was discovered that treatment using TP alone significantly decreased *Cyclin d1* and *c-Myc* mRNA levels and reduced their respective protein expression the most. This indicates that TP may suppress the expression of *Cyclin d1* and *c-Myc* mRNA while also promoting the degradation of these proteins (Figure 3C). 

### 2.4. TP Reduces the Body Length of Zebrafish, Similar to the Combined Treatment with P6 and DAPT

Although TP has broad antitumor effects, its clinical use is limited due to its poor water solubility and some toxic effects [2,42]. Therefore, before exploring TP as an anticancer agent, we confirmed its chemical toxicity and side effects using a zebrafish model system. To compare the results of experiments conducted at the cellular level with those observed in zebrafish, embryos were exposed to each inhibitor at 24 h post-fertilization (1 dpf), which is the stage when the embryos naturally hatch from the chorion. The zebrafish were sampled at 4 days post-fertilization (4 dpf) after the completion of the developmental stage. The experiment measured the mRNA levels of the target genes downstream of each signal, as shown in Figure 4A. The results of the experiment were similar to the experiment on HeLa cells (Figure 1). When only treated with P6, a STAT3 signaling inhibitor, zebrafish *Survivin* mRNA decreased. Similarly, when treated with DAPT, a Notch1 signaling inhibitor, the number of zebrafish *Notch1* mRNA decreased. Zebrafish *Cyclin d1* and *c-Myc* mRNA amounts were also reduced when treated with each inhibitor individually, indicating that they are common downstream targets of STAT3 signaling and Notch1 signaling (Figure 4A). When both inhibitors were applied simultaneously, the mRNA reduction for all downstream targets tended to be slightly greater than when P6 or DAPT were used individually. Furthermore, the reduction effect was more pronounced with TP treatment. The impact of reducing the common target mRNA during TP treatment was also confirmed by a decrease in the proteins c-Myc and Cyclin D1 (Figure 4B).

We conducted an experiment to assess the impact of different signaling inhibitors, including TP, on the developmental stages of zebrafish. We treated 100 zebrafish similarly and collected samples at 4 days post-fertilization (dpf). We then measured the body length of each zebrafish. Measuring the body length of zebrafish is a simple experimental method used to test the toxicity of TP and effectively observe the muscle formation process simultaneously [28,29]. This is a standard method in the zebrafish system and is widely used as an animal model for various muscle diseases [43,44].

When we treated the inhibitors under the same conditions as the previous experiment, we found that zebrafish treated with either P6 or DAPT did not show any significant difference in body length. Zebrafish treated with both P6 and DAPT showed a similar reduction in body length as zebrafish treated with TP, although the extent of the reduction was smaller (Figure 4C). These results are consistent with the data from previous cell viability assays (Figure 2), indicating that neither STAT3 nor Notch1 signaling alone significantly affects zebrafish body length changes. In the experiment, body length decreased when both signals were blocked simultaneously or when TP was administered. TP treatment was particularly effective in reducing body length.

### 2.5. TP Impairs Zebrafish Muscle Development

Based on the experiments conducted, it has been observed that blocking both STAT3 and Notch1 signaling simultaneously can reduce the body length of zebrafish (Figure 4C). A detailed observation of these samples using photos captured by a confocal fluorescence microscope revealed significant changes between the 11th and 14th somites in zebrafish. (Figure 5A). 

The expression patterns of muscle markers F-actin and myosin heavy chain (HC) were found to have been altered. F-actin an myosin HC are proteins expressed in muscle fibers and somites, helping monitor muscle development and differentiation [45,46]. No significant changes were observed in the expression of F-actin and myosin HC in samples where P6 or DAPT alone was applied compared to the control group. However, when samples were treated with both P6 and DAPT, the relative expression of F-actin and myosin HC decreased, leading to empty spaces in the somites. The arrangement of muscle fibers appeared irregular. The application of TP further aggravates this phenomenon and disrupts the regular expression pattern of each marker protein (Figure 5A). We have noticed the same effect on C2C12 mouse myoblasts when treated with TP, as shown in Figure 5D. When we induced differentiation using DM after TP treatment, the expression of myosin HC decreased in comparison to the control group. We measured the survival rate of C2C12 cells after TP treatment before differentiating them and found that TP inhibited muscle cell proliferation, as it does with other cells. This fact demonstrates that TP is ultimately involved in muscle fiber formation, which is a critical step in myogenesis, through its effect on cell viability. This is thought to be due to decreased F-actin and myosin HC expression or abnormally rapid degradation of muscle marker proteins during muscle formation. After obtaining the results, we investigated whether the signaling of STAT3 or Notch1 affects the expression of the myogenic regulatory factors MyoD and myogenin, and Pax7, a transcription factor for muscle cell proliferation [47,48,49]. Our findings showed that TP treatment significantly reduced MyoD, myogenin, and Pax7 proteins under conditions that led to a decrease in p-STAT3 due to the inhibition of STAT signaling and a reduction in Hes1 protein, a direct signaling target of Notch1 (Figure 5B). However, the simultaneous application of P6 and DAPT resulted in a smaller decrease in these proteins compared to TP treatment (Figure 5B). We compared a negative control with TP-treated zebrafish and observed the most severe phenotypic defects. The TP-treated zebrafish showed unusual body shapes, such as large yolk sacs, bent tails, and axis shortening, in addition to variations in body length, segmentation, and muscle fiber arrangement (Figure 5C).

## 3. Discussion

We began our study by screening 1100 natural and FDA-approved compounds, with the goal of discovering new inhibitors of the Notch signaling pathway [50]. Notch1 signaling is responsible for regulating cell differentiation, proliferation, and apoptosis, and its abnormal activation has been linked to the development and progression of various cancers [27,50,51,52,53,54,55,56]. Using a luciferase reporter assay, we identified 14 compounds that successfully inhibited Notch signaling (Appendix A). Among these, we focused on TP, a natural product, and confirmed its ability to inhibit Notch1 signaling (Appendix A). According to our findings, TP can regulate Notch signaling and impact different downstream genes, leading to various physiological alterations. There are reports that TP can hinder the protein expression of Notch1 and its active form NICD, which in turn reduces PTEN, causing oxidative stress in hepatocytes and liver damage [57]. In addition, TP can prevent the accumulation of extracellular matrix (ECM) proteins and inhibit Notch1 pathway activation in diabetic conditions [58]. A possible mechanism for the regulation of Notch signaling by TP is the inhibition of the enzymatic activity of Adam10, a metalloproteinase involved in the initial cleavage of Notch, through the direct binding of TP to Adam10 [59,60].

To determine whether TP could inhibit Notch signaling as effectively as DAPT, a γ-secretase inhibitor (a specific Notch signaling inhibitor), we examined the expression of Notch1 and Hes1 mRNA (Appendix A) and the levels of downstream targets of Notch signaling, Hes5 and NICD (Appendix A). Our results demonstrated that TP was indeed able to inhibit Notch signaling when used at the same concentration as DAPT. Interestingly, while DAPT was unable to inhibit the proliferation of HeLa cells, TP was able to do so at the same dose (Appendix A).

These findings suggest that inhibiting Notch1 signaling alone is insufficient to achieve a practical anti-proliferative effect, and that the additional regulation of another signal is required. Our investigation focused on signaling pathways commonly involved in the diverse anticancer effects of TP [4,6]. Previous studies have shown that TP can exert anticancer effects by down-regulating STAT3 signaling in various cancer cells, such as ovarian [11], colon [13], lung [61], and multiple myeloma cells [12]. Thus, we had a hypothesis that TP possesses potent anti-prolierative properties by regulating Notch1 and STAT3 signaling simultaneously. To test this hypothesis, we conducted experiments using DAPT and P6, which are known signaling inhibitors. Our findings show that TP is capable of inhibiting Notch1 and STAT3 signaling pathways that are responsible for regulating cell fate. Interestingly, the use of DAPT and P6 inhibitors together had a slightly weaker anti-proliferative effect than TP alone. This suggests that TP may be regulating other essential signals besides Notch1 and STAT3.

The active compound found in TWHF, TP, is known for its anti-inflammatory effects. Studies have shown that 10–100 femtomolar of TP, which can be extracted by boiling tea from the plant, can help prevent inflammation and muscle atrophy [62]. However, this study found that TP caused abnormalities in muscle cells due to its anti-proliferative activity. This might be due to differences in treatment concentration. When treated with tens of fM concentrations, there was no anti-proliferative effect on cancer cells (Appendix A).

In this study, we explored the possibility of using TP as an anticancer agent and assess its side effects at concentrations ranging from tens to hundreds of nanomolar (nM), the concentration level when applying commonly known anticancer drugs [63,64]. Chemotherapy at this concentration may cause muscle loss in cancer patients due to drug-related adverse reactions or complex mechanisms [65].

Skeletal muscle development is a complex process that involves maintaining a balance between catabolic and anabolic processes, controlled by many muscle-related genes and their transcription factors. Studies have shown that an abnormal regulation or inhibition of essential signals in this process can lead to skeletal muscle atrophy, where protein breakdown increases instead of protein synthesis [62].

The shift towards catabolic processes can inhibit PTEN/Akt and NF-kb signaling, which can increase protein degradation through ubiquitin–proteasome-mediated proteolysis and autophagy-lysosomal systems, ultimately leading to muscle loss [57,66]. 

Satellite cells are a group of stem cells that maintain skeletal muscle tissue throughout life. These cells are responsible for muscle growth, repair, and regeneration, and are crucial for skeletal muscle function [30,67]. During adult muscle development, the progression of myoblast muscle precursor cells originating from satellite cells is regulated by the balance between Notch and Wnt signaling [68].

Notch1 and JAK1-STAT3 signaling increase the expression of MyoD and myogenin, which are critical myogenic regulatory factors of muscle development, thereby facilitating myoblast proliferation [47,69,70]. These transcription factors control the proliferation, differentiation, and fusion of muscle cells by following a specific sequence of expression [48,49,67]. 

These transcription factors play a crucial role in increasing muscle mass by promoting myoblasts proliferation. They also help in rearranging the existing muscle fibers after birth. In addition, they stimulate the proliferation of muscle stem cells in case of muscle damage due to various reasons [30,31]. That is, transcription factors are responsible for postnatal muscle development and maintenance, so they also participate in epigenetic regulation that contributes to final muscle maintenance [71]. 

It was reported that the downregulation of myoD and myogenin activates protein degradation, leading to muscle weakness and atrophy [71]. It seems that the reduction in the protein levels of transcription factors required for muscle growth may have resulted in muscle atrophy by failing to properly enhance the muscle cells.

Our experiments on zebrafish muscle development showed that the levels of two specific proteins, MyoD and myogenin, were reduced while undergoing TP treatment. Additionally, the level of Pax7, which plays a crucial role in the initial stages of myoblast proliferation [49], was also decreased (Figure 5B). 

Notably, Figure 5C shows that zebrafish treated with TP had unusual body shapes, including large yolk sacs, curved tails, and axial shortening. TP-treated zebrafish were also changed in body length, segmentation, and muscle fiber arrangement. These changes could potentially cause developmental problems and hinder the absorption of nutrients from the yolk. This process is typical of an inflammatory response caused by liver damage. It is known that TP can cause hepatotoxicity and affect liver structure by activating apoptosis and autophagy, leading to liver damage [72]. It is known that pro-inflammatory cytokines induce signals that activate muscle weakness through ubiquitin–proteasome-mediated proteolysis [71]. Although not covered in this experiment, TP can probably also affect muscle structure and induce muscle loss through the exact mechanism. 

All these facts suggest that TP may interfere with muscle development in zebrafish by simultaneously suppressing STAT3 and Notch1 signaling and influencing other signals to induce various phenotypes, ultimately leading to muscle atrophy.

Muscle atrophy occurs due to an imbalance in protein synthesis and breakdown, caused by various factors like aging, oxidative stress, and inflammation. This experiment aimed to examine the expression of proteins that are essential for muscle homeostasis and maintenance. We studied the changes in muscle-related proteins during the inhibition of muscle cell proliferation by TP throughout the zebrafish muscle development process. Our objective was to determine whether these proteins are involved in muscle atrophy. However, the study only provided one possible cause, and no research was conducted on the relationship between TP and the actual protein breakdown process involved in adult muscle atrophy. Furthermore, it is widely known that TP has an anti-inflammatory effect which is helpful in preventing muscle atrophy caused by inflammation. However, there are contradictory results regarding the fact that TP, as an anticancer substance, can also induce cancer cachexia. This aspect seems to be related to the concentration of treatment, but the study did not investigate this. Future research should investigate this area separately. Nonetheless, this study identified that at least two critical signals are involved in the mechanism of skeletal muscle atrophy. 

Our studies suggest that TP is effective against cancer, but it can also cause muscle loss by degrading proteins or by reducing the expression of downstream targets that are involved in muscle development. The relationship between TP and muscle loss is illustrated in Figure 6, which demonstrates that TP regulates Notch1 and STAT3 signaling to impact cell proliferation, muscle development, and homeostasis. Ultimately, this contributes to skeletal muscle atrophy.

## 4. Materials and Methods

### 4.1. Antibodies

Antibodies were used in Western blotting and immunohistochemistry (IHC). The primary antibody for NICD (#07-1231, 1:1000) was purchased from EMD Millipore Corporation (EMD Millipore Corporation, Temecula, CA, USA). The primary antibodies for c-Myc (#18583, 1:1000), Survivin (#2808, 1:1000), Cyclin D1 (#55506, 1:1000), and Bcl-2 (#15071, 1:1000), were purchased from Cell Signaling (Cell Signaling Technology, Beverly, MA, USA). The primary antibody for Pax7 (#CSB-PA017493ESR1HU, 1:1000) was purchased from Cusabio (Cusabio, Wuhan, China). The primary antibodies for p-Stat3 (#sc-136193, 1:1000), β-actin (#sc-47778, 1:5000), and HES1 (#sc-166410, 1:1000) were purchased from Santa Cruz (Santa Cruz Biotechnology, Santa Cruz, CA, USA). The primary antibody for MyoD (#MA5-12902, 1:1000) was purchased from Invitrogen (Grand Island, NY, USA). The primary antibody for myogenin (#A17427, 1:2000) was obtained from ABclonal (ABclonal technology, Woburn, MA, USA). For IHC, the primary antibody for myosin heavy chain (#MAB4470, 1:5000) was purchased from R&D Systems (a Bio-Techne Brand, Minneapolis, MN, USA). 

### 4.2. Cell Maintenance

HeLa cells, L929 cells, U251 cells, and Jurkat cells were purchased from the Korean cell line bank (Seoul, Republic of Korea) and used for the following experiments. HeLa (cervical cancer cell), L929 (murine skin fibroblasts), and U251 (glioma cell) cells were cultured in DMEM (Welgene, Gyeongsan-si, Republic of Korea), supplemented with 1% antibiotic antimycotic solution, 100× (Corning, Manassas, VA, USA) and 10% fetal bovine serum (Welgene, Gyeongsan-si, Republic of Korea). Jurkat (T-Cell Lymphocyte leukemia) cells were cultured in RPMI (Welgene, Gyeongsan-si, Republic of Korea), supplemented with 10% FBS and 1% Abs. All cells above were cultured in a 5% CO_2_ chamber at 37 °C. C2C12 (mouse myoblasts) cells were a gift from Mi-Ock Lee at Seoul National University in Seoul, Republic of Korea. C2C12 cells were cultured in DMEM supplemented with 10% FBS and 1% antibiotic antimycotic solution, 100× (Biowest, Riverside, MO, USA). For the induction of differentiation, fully confluent C2C12 cells were cultured in DMEM supplemented with 2% horse serum (Sigma-Aldrich, St. Louis, MO, USA) for 3 days.

### 4.3. Zebrafish System

Several lines of wild type of Zebrafishes and automatic water circulation, temperature control systems (Genomic-Design, Daejeon, Republic of Korea) were kindly provided by Dr. Jieun Lee (SKKU, Seoul, Republic of Korea). Maintaining the zebrafish system was completed according to the reference [73]. Here is a brief explanation of the contents. After 6 months post-fertilization, wild-type zebrafish (*Danio rerio*) were kept in a 14 h light cycle in the system’s water at 28.5 ℃. Zebrafish were raised in several individual tanks and circulated system water through the central tank. The circulating water is purified with several (biological, active carbon, etc.) filters (S&F, Daejeon, Republic of Korea). The temperature of the fish room and system water was controlled using air-conditioning and auto-water heaters. Embryos were collected from our wild-type fishes and raised in E3 buffer with 0.001 M methylene blue (Sigma-Aldrich, St. Louis, MO, USA), which was used to prevent bacterial infections. Animal subject research was reviewed and approved by the Institutional Animal Care and Use Committee at the Sejong University (SJ-20211115).

### 4.4. Chemical Treatment

Natural compounds (#L1400) and FDA-approved chemicals (#L1300), including triptolide and DAPT for the Notch inhibitor screening test, were purchased from Selleckchem, (Houston, TX, USA). DMSO was purchased from Corning (USA, Manassas, VA, USA). Pyridone 6 (P6) was purchased from R&D Systems (a Bio-Techne Brand, Minneapolis, MN, USA). In cell-based experiments, P6, DAPT, P6 + DAPT, TP, or DMSO were treated for a selective concentration of 50 nM, 100 nM, 200 nM, 500 nM. The concentrations of each signal inhibitor and TP were determined through referencing their use in various cells [39,40,41] and zebrafish [74,75]. The concentration ranges typically used in cell-based experiments to test anticancer agents was applied [76]. They were incubated at 37 °C, 5% CO_2_, for 24 h. To reduce error, chemicals were diluted with Opti-MEM (Welgene, Republic of Korea) before suspension. In zebrafish experiments, 24 hpf embryos were placed in 90ϕ Petri dishes filled with 10 mL of E3 buffer (with 0.001 M methylene blue). Embryos were treated with DMSO for the negative control or a 500 nM concentration of P6, DAPT, P6 + DAPT, or triptolide. They were incubated for 72 h in an incubator at 28.5 °C.

### 4.5. Luciferase Reporter Assay

Our lab previously made an inducible NICD-expressing stable cell line (HeLa RBP-JK (Luc/Ren), through infecting HeLa cells through Lentivirus [50]. Cells were grown to about an 80% cell concentration in 96-well plates and treated with several compounds for 24 h. Then, the cells were lysed in 5× Passive Lysis Buffer (Promega, Madison, WI, USA) and dual-analyzed for Luciferase and Renilla activity with a GloMax^®^ 96-well microplate luminometer system (Promega). The Luciferase reporter activity in each sample was normalized with Renilla protein activity (Promega).

### 4.6. Quantitative RT-PCR (qRT-PCR)

Trizol reagent (Life Technologies, Carlsbad, CA, USA) was used for the total RNA isolation of the cells and zebrafish. Zebrafish embryos were needed to homogenize in Trizol reagent for total RNA isolation. A total of 1 μg RNA was used with random primer mix for cDNA synthesis (Intron Bio, Seongnam, Republic of Korea). cDNA was amplified using primer pairs for human *STAT3* (forward 5′-TTGACAAAGACTCTGGGAC-3′, Reverse 5′-CAGGGAAGCATCACAATTGG-3′), human *Notch1* (forward 5′-TACGTGTGCACCTGCCGGG-3′, reverse 5′-CGTTTCTGCAGGGGCTGGGG-3′), human *Bcl2* (forward 5′-GTGGCCTCTAAGATGAAGGA-3′, reverse 5′-TGCGGATGATCTGTTTGTTC-3′), human *Cyclin d1* (forward 5′-GGATTGTGGCTTCTTTGAGGA-3′, reverse 5′-AGGTACTCAGTCATCCACAG-3′), human *c-Myc* (forward 5′-GCTGCTTAGACGCTGGATTT-3′, reverse 5′-CACCGAGTCGTAGTCGAGGT-3′), human *Survivin* (forward 5′-CCTTCACATCTGTCACGTTC-3′, reverse 5′-GAAGCTGTAACAATCCACCC-3′), human *Hes1* (forward 5′-ATGACGGCTGCGCTGAGCAC-3′, reverse 5′-TAACGCCCTCGCACGTGGAC-3′), human *Hes5* (forward 5′-CCGGTGGTGGAGAAGATG-3′, reverse 5′-GACAGCCATCTCCCAGGATGT-3′), human *β-actin* (forward 5′-GAGACCTTCAACACCCCAGCC-3′, reverse 5′-GGATCTTCATGAGGTAGTCAG-3′), zebrafish *Notch1* (forward 5′-TTCTGGCATTCACTGTGAGC-3′, reverse 5′-TCTCTCTGTCCTGGCAGGTT-3′), zebrafish *Bcl2* (forward 5′-GGAAATGTCCCAACAAATGG-3′, reverse 5′-TATCTACCTGGGACGCCATC-3′), zebrafish *Survivin* (forward 5′-CGTTTGCACTCCAGAAAACA-3′, reverse 5′-AATCACAGCTGGGAGAATGC-3′), zebrafish *c-Myc* (forward 5′-AGAGAGACTGGCGTCTTTGC-3′, reverse 5′-GCTGGAGCTGTTAGGTGGAG-3′), zebrafish *Cyclin d1* (forward 5′-CTGTGCGACAGACGTCAACT-3′, reverse 5′-CTGACA CGATCGCAGACAGT-3′), and zebrafish *β-actin* (forward 5′-CGAGCAGGAGATGGGAACC-3′, reverse 5′-CAACGGAAACGCTCATTGC-3′). The primer pairs were purchased from Cosmogenetech (Seoul, Republic of Korea). A Step One Real-Time PCR System was used for conducting qRT-PCR. cDNA was amplified using the primers described above and a Real-Time PCR 2× Master Mix-SYBR green (Elpis-Biotech, Daejeon, Republic of Korea), according to the manufacturer’s protocol.

### 4.7. Western Blotting

Cells were grown to about 80% cell concentration in 6-well plates and treated with selective chemicals for 24 h. They were lysed in 5X Passive Lysis Buffer (Promega) with Xpert Protease Inhibitor cocktail solution (GenDepot, Katy, TX, USA). The supernatant was transferred to a fresh tube for the protein sample and the remaining pellet was discarded. Then, a BCA protein assay (Thermo Fisher Scientific, Waltham, MA, USA) was conducted to measure the protein concentration. In the zebrafish experiments, approximately 60–70 of the 24 hpf embryos were placed in each well of the 6-well plates. They were homogenized through a 1 mL syringe in 200 μL of RIPA buffer with Xpert Protease Inhibitor cocktail solution (GenDepot, Katy, TX, USA) and incubated in ice for 20 min. The supernatant was transferred to a fresh tube for the protein sample and the remaining pellet was discarded. Then, a BCA protein assay (Thermo Fisher Scientific) was conducted to measure the protein concentration. For Western blotting, all the protein samples were separated using SDS-PAGE (Polyacrylamide gel electrophoresis) and semi-dry transferred into PVDF (Polyvinylidene difluoride) membranes (Milipore, Billerica, MA, USA). The PVDF membranes were put into 5 mL of 5% skim milk solution from TBS with 0.5% Tween-20 for 1 h at R.T. for blocking. The membranes were incubated with a selective primary antibody in blocking solution and placed on a bench rocker at 4 °C overnight. The membranes were then rinsed 3 times with 5 mL of 0.5% TBST and placed on a bench rocker at R.T. for 5 min each. After removing the last 0.5% TBST, the membranes were incubated with a selective secondary antibody conjugated with HRP in block solution and placed on a bench rocker at R.T. for 1 h. After washing 3 times with 0.5% TBST, the membranes were developed using the ECL (enhanced chemiluminescence) detection system (Bio-Rad, Hercules, CA, USA).

### 4.8. Cell Viability Assay

MTT assay and CCK-8 assay were performed to measure cell viability. A MTT cell viability assay kit (Sigma-Aldrich, St. Louis, MO, USA) and a Cellrix viability assay kit (MediFab, Seoul, Republic of Korea) were used, following the manufacturer’s instruction. To measure absorbance, which directly correlates to the quantities of viable cells, a Model 680 microplate reader (Bio-Rad, Hercules, CA, USA) was used. 

### 4.9. Measurement of Zebrafish Body Length

Embryos were placed in 100ϕ Petri dishes and treated with DMSO for the negative control or a 500 nM concentration of P6, DAPT, P6 + DAPT, TP at 24 h post-fertilization (hpf). Treated embryos were stunned with Tricaine and placed on sample slide at 3 days post-emulsion (dpe). Then, one hundred embryos per each sample were measured using a SMZ1000 stereomicroscope (Nikon Inc, Tokyo, Japan).

### 4.10. Immunohistochemistry

Embryos placed in 100ϕ Petri dishes were transferred into 12-well plate. Approximately 10–20 embryos were put into each well of the 12-well plate. Then, 1 mL of 4% paraformaldehyde (Sigma-Aldrich, MO, USA) was added to each well for embryo fixation and placed on a bench rocker at 4 °C overnight. The fixed embryos were rinsed 3 times with 1 mL of PBS (Gibco, Thermo Fisher, Waltham, MA, USA) with 0.5% Tween-20 (Bio-Basic, Markham, ON, Canada), and placed on a bench rocker for 5 min each. After removing the last 0.5% PBST, 1 mL PBS with 2% Triton^®^X-100 (USB corporation, Cleveland, OH, USA) was added on each well for 2 h at RT to permeabilize the embryos for phalloidin staining. The PBS 2% TritonX-100 was removed and each well was refilled using fresh 500 mL PBS 2% TritonX-100 with Alexa Fluor^®^488 phalloidin (provided by Dr. Jieun Lee, SKKU, Seoul, Republic of Korea), conjugated with an F-actin probe. A 12-well plate was wrapped in foil and was placed on a bench rocker at 4 ℃ overnight. After this, every stage of samples was wrapped in foil. Before adding primary antibody solution, phalloidin stained embryos were rinsed 3 times with 1 mL of 0.5% PBST and placed on a bench rocker for 5 min each. Then, the embryos were blocked with 2 mL of 2% BSA for 2 h at RT. When blocking was completed, the myosin heavy chain primary antibody was added into the blocking solution (1:5000) and placed on a bench rocker at 4 °C overnight. The embryos were washed 3 times with 1 mL of 0.5% PBST and placed on a bench rocker for 5 min each. After removing the last washing solution, 2 mL of 2% BSA with Alexa Fluor^®^594 conjugated secondary antibody (1:5000) was added. The embryos were washed with 1 mL of 0.5% PBST as needed and 2 mL of 2% BSA with DAPI solution (#10236276001, 1:5000, Roche Diagnostics, GmbH, Germany) was added. Embryos were washed with 1 mL of 0.5% PBST as needed. The embryos were rinsed with PBS buffer several times. After removing the last 0.5% PBS, each well was filled with PBS:Glycerol (4:6) and kept it in refrigerator at 4 °C. The PBS:Glycerol (2:8) was used for mounting the solution and to help the mounted sample to fit under the microscopes.

### 4.11. Statistical Analysis

All the data were presented as the mean ± standard deviation (S.D). For the qRT-PCR and measurement of body length for the zebrafish, the statistical significance was determined with the Student *t*-test with a significance level of *, *p* < 0.05 or **, *p* < 0.01. The data for the qRT-PCR, the viability assay, and the Western blot were normalized using the negative control (DMSO) and presented as the mean of three independent experiments in triplicate. For the Western blot, ImageJ software (https://imagej.net/software/fiji/downloads, accessed on 21 August 2023, NIH, Bethesda, NY, USA) was used to graphically display the protein levels for ease of understanding.

## 5. Conclusions

Our research suggests that TP can hinder the growth of cancer cells by inhibiting the signaling pathways of Notch1 and STAT3. However, it can also lead to muscle loss by activating proteolysis, which results from the suppression of the expression of myoD, Pax7 and myogenin. These two transcription factors play a critical role in muscle development and maintaining muscle mass. The findings of this study suggest that chemotherapy can cause muscle atrophy as a side effect, due to a common mechanism in which various anticancer drugs inhibit Notch1 and STAT3 signaling, similar to TP treatment. In conclusion, our study highlights that TP, an effective cancer treatment, may cause muscle loss by significantly decreasing the expression of target genes responsible for muscle development and maintenance.

## Figures and Tables

**Figure 1 ijms-25-04675-f001:**
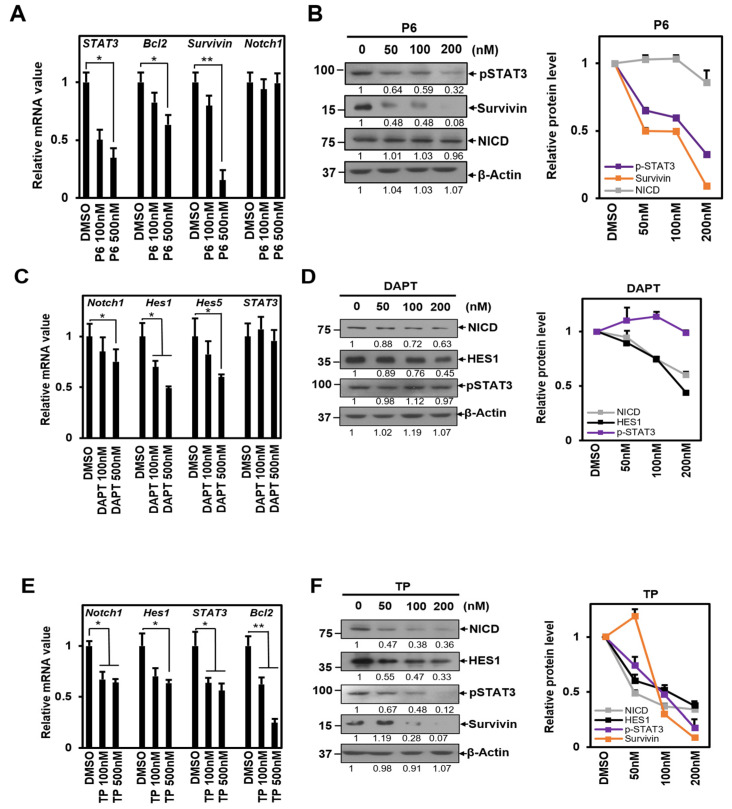
TP inhibits both STAT3 and Notch1 signal. Hela cells were treated with 50 nM, 100 nM, 200 nM, 500 nM concentration of (**A**,**B**) P6, (**C**,**D**) DAPT, (**E**,**F**) TP or control (DMSO) for 24 h. Total RNA was isolated and subjected to qRT-PCR analysis. Data were normalized to β-Actin expression. Treated cells were lysed and subjected to Western blotting with antibody against p-STAT3, Survivin, NICD, HES1, and β-Actin. Membranes were analyzed using ImageJ software (NIH, Bethesda, NY, USA), and changes in protein levels were graphically displayed next to each other for ease of understanding. The results represent the means ± S.D. of three independent experiments performed in triplicate. *, *p* < 0.05; **, *p* < 0.01. P6: Pyridone 6, a pan Janus kinase inhibitor; DAPT: γ-secretase inhibitor; TP: triptolide.

**Figure 2 ijms-25-04675-f002:**
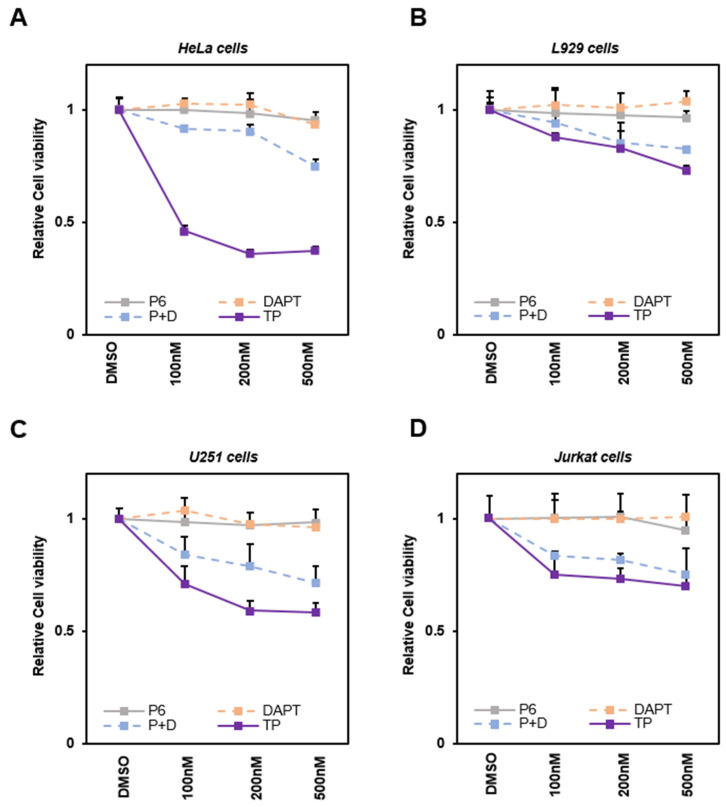
TP reduces the viability of various cancer cells. (**A**) HeLa cells, (**B**) L929 cells, and (**C**) U251 cells were treated with control (DMSO) or respective concentrations of 100 nM, 200 nM, and 500 nM for P6, DAPT, P6 + DAPT, and TP. Cells were cultured for 48 h. Cell viability was measured using MTT assay in 24-well plates. (**D**) Viability of Jurkat suspension cells was measured using CCK-8 assay in 24-well plates. P + D refers to simultaneously applied P6 and DAPT. The results represent the means ± S.D. of three independent experiments performed in triplicate. P6: Pyridone 6, a pan Janus kinase inhibitor; DAPT: γ-secretase inhibitor; TP: triptolide.

**Figure 3 ijms-25-04675-f003:**
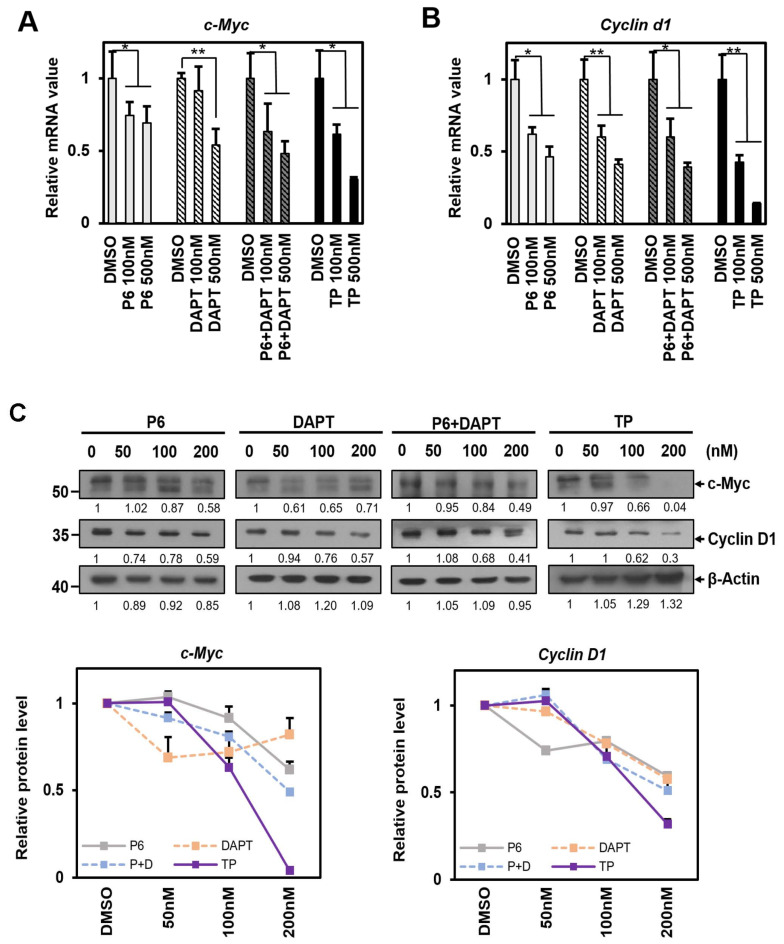
TP reduces *c-Myc* and *Cyclin d1* mRNA and these proteins in cell proliferation. (**A**,**B**) HeLa cells were treated with control (DMSO) or P6, DAPT, P6 + DAPT, and TP at the indicated concentrations for 24 h. Cells were harvested and total RNA was isolated and subjected to qRT-PCR analysis. Data were normalized to β-Actin expression. (**C**) Treated cell lysates were subjected to Western blotting with antibodies against c-Myc, Cyclin D1, and β-Actin. Protein levels were graphically displayed below for ease of understanding using ImageJ software (NIH, Bethesda, NY, USA). The results represent the means ± S.D. of three independent experiments performed in triplicate. *, *p* < 0.05; **, *p* < 0.01. P6: Pyridone 6, a pan Janus kinase inhibitor; DAPT: γ-secretase inhibitor; TP: triptolide.

**Figure 4 ijms-25-04675-f004:**
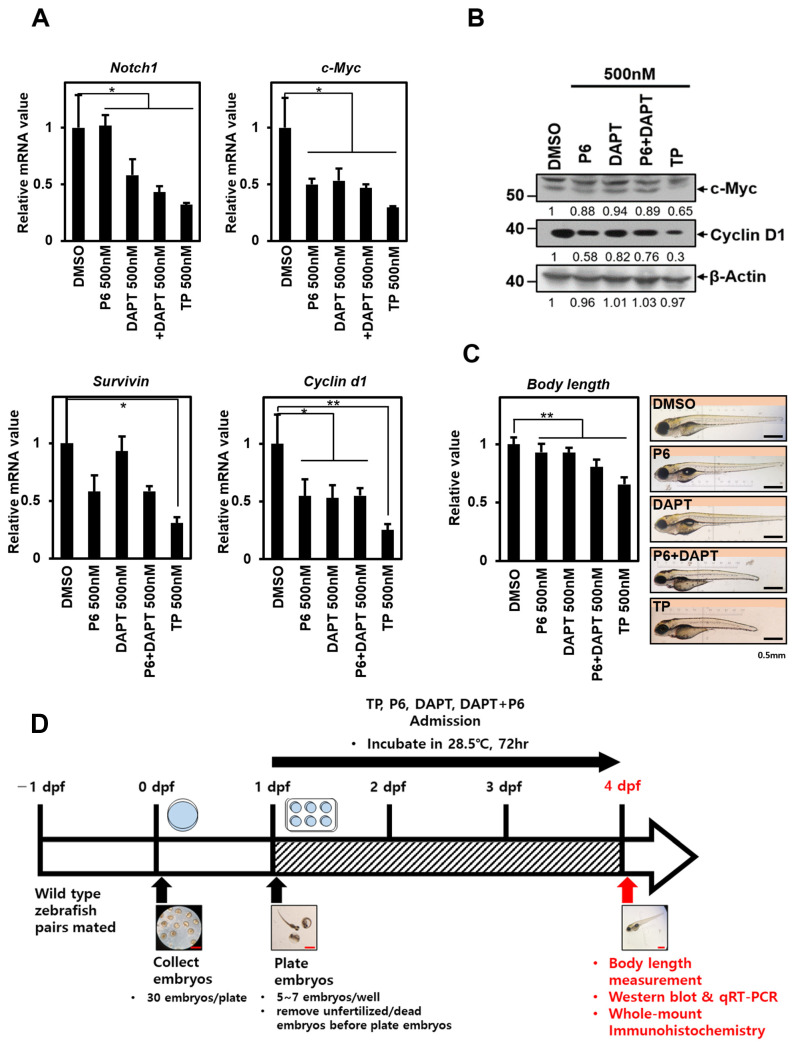
TP reduces the body length of zebrafish, similar to the combined treatment with P6 and DAPT. Embryos were treated with control (DMSO) or 500 nM concentration of P6, DAPT, P6 + DAPT, and TP at 1 dpf. (**A**). Embryos were homogenized at 4 dpf and subjected to qRT-PCR analysis targeting *Notch-1*, *c-Myc*, *Survivin*, and *Cyclin D1*, which are jointly affected by STAT3 and Notch1 signaling. Data were normalized to β-Actin expression. The results represent the means ± S.D. of three independent experiments performed in triplicate. *, *p* < 0.05; **, *p* < 0.01. (**B**) Homogenized embryos were lysed and subjected to Western blot using antibodies against c-Myc, Cyclin D1, and β-Actin. Membranes were analyzed using ImageJ software (NIH, Bethesda, NY, USA). (**C**) Pictures of treated larvae were taken at 4 dpf, after stunning with Tricaine. The pictures were taken using an SMZ1000 stereomicroscope. On the bottom right side of the images is a black line indicating 0.5 mm, 1/4 length of the ruler in the microscope. The graph next to the pictures shows the average body length of 100 embryos. (**D**) Diagram depicting the overview of zebrafish experimental process in sequence. Red scale bars: 0.5mm; dpf: day-post fertilization; P6: Pyridone 6, a pan Janus kinase inhibitor; DAPT: γ-secretase inhibitor; TP: triptolide.

**Figure 5 ijms-25-04675-f005:**
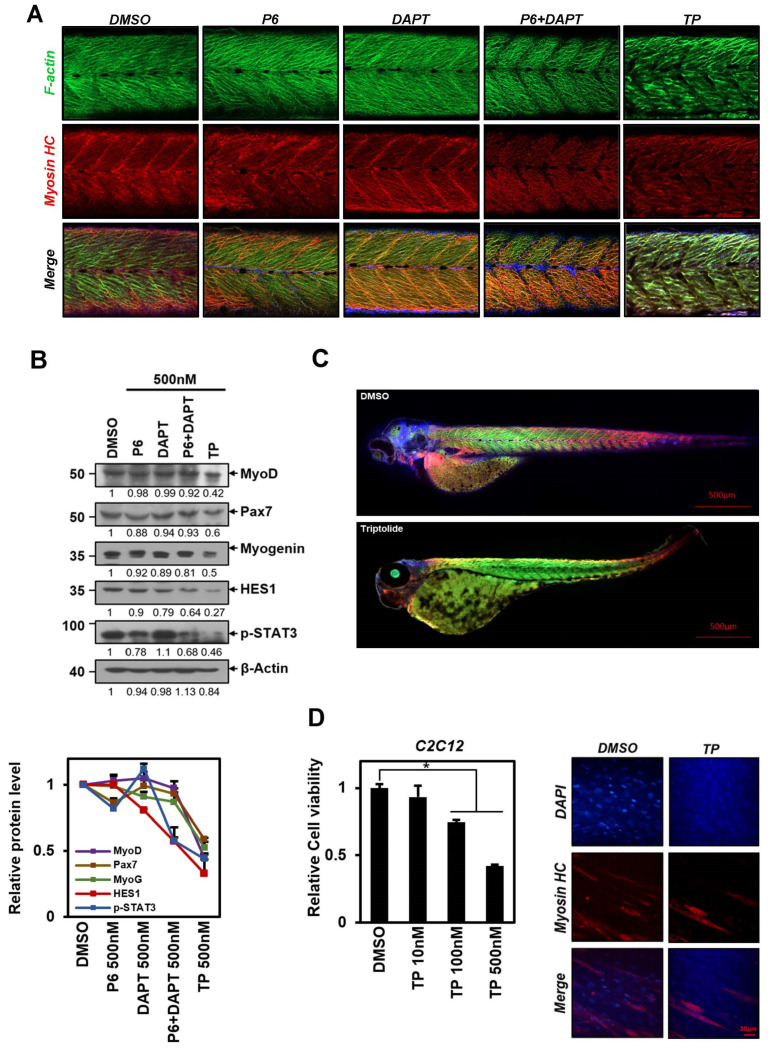
TP impairs muscle development in zebrafish by inhibiting muscle cell proliferation. (**A**) Zebrafish embryos were treated with control (DMSO) or 500 nM concentrations of P6, DAPT, P6 + DAPT, and TP at 24 hpf, then fixed at 3 dpf and incubated with anti-F-actin conjugated phalloidin (green), anti-myosin heavy chain (red), and DAPI. (blue). The images shown in the whole-mount immunohistochemistry experiments represent the enlarged image of 8th to 11th somites, as shown in Figure 5C. (**B**) Treated embryos were lysed and subjected to Western blot using antibody against MyoD, Pax7, Myogenin, Hes1, pSTAT3 and β-Actin. Protein levels were graphically displayed below for ease of understanding using ImageJ software (NIH, Bethesda, NY, USA). (**C**) The pictures show full-size DMSO (control) or TP-treated embryos subjected to whole-mount IHC. (**D**) C2C12 cells were exposed to TP for 24 h, and their viability was assessed using the MTT assay in 96-well plates. After treating C2C12 cells in a 6-well plate with TP for 24 h, the medium was replaced with differentiation media (DM). The results represent the means ± S.D. of three independent experiments performed in triplicate. *, *p* < 0.05. At day 3 of differentiation, immunocytochemistry images were obtained using anti-MYHC antibodies to stain the TP (500 Nm)-treated or DMSO-treated C2C12 cells. The nucleus was visualized using DAPI (blue). Scale bar = 30 μm. P6: Pyridone 6, a pan Janus kinase inhibitor; DAPT: γ-secretase inhibitor; TP: triptolide; Myosin HC: myosin heavy chain.

**Figure 6 ijms-25-04675-f006:**
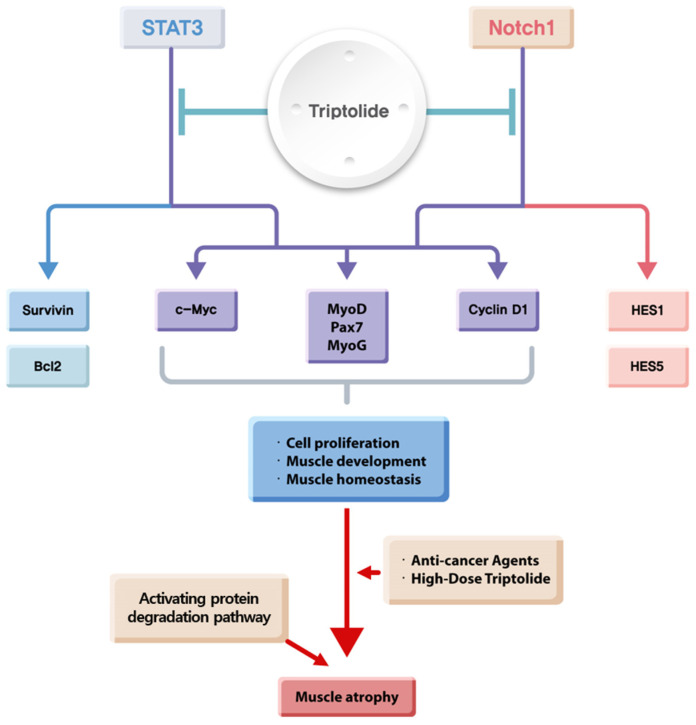
Diagram describing the inhibitory role of TP in STAT3 and Notch1 signaling in cell proliferation and muscle development and homeostasis. TP simultaneously inhibits essential signaling pathways, including Notch1 and STAT3 signaling, thereby inhibiting cancer cell proliferation and causing muscle atrophy by regulating the expression of common targets such as *c-Myc*, *Cyclin D1*, *MyoD*, *Pax7* and *MyoG*, and contributing to inducing muscle atrophy upon activation of protein degradation. MyoG: myogenin.

## Data Availability

Available in section “MDPI Research Data Policies” at https://www.mdpi.com/ethics (accessed on 6 March 2024).

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
