# Peer review of "Triptolide, a Cancer Cell Proliferation Inhibitor, Causes Zebrafish Muscle Defects by Regulating Notch and STAT3 Signaling Pathways"

_ijms, 2024, doi:10.3390/ijms25094675_

Round 1

Reviewer 1 Report (Previous Reviewer 2)

Comments and Suggestions for Authors

This article possesses slight originality, but would match this special issue.

Author Response

Dear reviewer #1

We thank you for your valuable feedback, which ensures that our manuscript will match this special issue with its originality. We have responded to two other reviewers' comments, as attached, and included additional experiments in this revision. We hope that these changes will lead to better results and address any of your concerns. Thank you for taking the time to carefully review our work.

Best regards,

Jaekyung Shim, Ph.D.

Assistant professor

Department of Bioresources Engineering, Sejong University,

209, Neungdongro, Kwangjin-Gu, Seoul 05006, Korea

Phone: 82-2-3408-3944 Fax: 82-2-3408-4318

E-mail: jkshim@sejong.ac.kr

Reviewer 2 Report (New Reviewer)

Comments and Suggestions for Authors

The articles “Triptolide, a cancer cell proliferation inhibitor, causes zebrafish muscle defects by regulating Notch and STAT3 signaling pathways” is aimed to find new anti-cancer and anti-inflamatory drugs. The authors performed the preliminary screen and identified triplolide (TP) as potent compound. It is of natural origin and approved by FDA for cancer treatment already. It is known to inhibit STAT and Notch signaling pathways which was confirmed in present article. Also, authors explored a range of cell lines to check the specificity of the drug to cancer cells. It turned out TP inhibits the abovementioned signaling pathways in all tested cells. Also, authors explored the side effect of the TP on muscle cells development in zebrafish model. And even though the effect of TP was similar somehow to that of double treatment with known inhibitors of Notch and STAT signaling pathways, it was much more severe.

The article is well-written and the research in described in details. However, some logic is lacking and needs more explanation: 1. Why MTT is compared with CKK-8 test? MTT evaluates the activity of mitochondria by measuring formazan crystals formation, unlike cell counting test that is evaluating the number of cells. And even though there is a link between cell count and physiological well-being of the cells, these are not the same. 2. Authors do not discuss the cross-talk of Notch and STAT3 with other signaling pathways. All the target genes might also be targets for other signaling pathways that play some role in cancer development, such as Wnt/beta-catenin. 3. The initial screening from the supplementary materials was done with much higher concentrations. Why use the concentration that works for other drugs? Are there IC50 comparable? 4. The test with zebrafish clearly indicates the toxicity of the TP. Why not to focus on other compounds? 5. What software was used to calculate statistics?

In conclusion, the article might be of interest to reader and bring some novel aspects of the TP role in cancer treatment, but authors need to explain more some aspects of their research and make it clear to the reader.

There also are some minor corrections:

Line 8 – Repeat of the word “correspondence”.

Line 174 – when you talk about genes, it is not genes interacting but products of genes.

Line 364 – paragraph missing.

Comments on the Quality of English Language

The English is clear and easy to understand, only minor corrections are needed.

Author Response

Dear reviewer #2

We would like to express our gratitude for your valuable feedback and suggestions. Your comments helped us improve our manuscript significantly in the revision. We have now made every effort to address your concerns more comprehensively in this revised version. We hope that our manuscript accurately reflects the reviewer's points and provides an appropriate response.

We have also attached our responses to other reviewers' comments to help you make decisions about our paper.

Thank you for taking the time to review our work carefully.

Best regards,

Jaekyung Shim, Ph.D.

Assistant professor

Department of Bioresources Engineering, Sejong University,

209, Neungdongro, Kwangjin-Gu, Seoul 05006, Korea

Phone: 82-2-3408-3944 Fax: 82-2-3408-4318

E-mail: jkshim@sejong.ac.kr

Reviewer 3 Report (New Reviewer)

Comments and Suggestions for Authors

I do not think this work is suitable for publication because the major drawback is that conclusions are not strongly supported experimental data. The manuscript does not provide mechanistic insights into the activity of triptolide in regulating gene expression.

1. The authors compared triptolide with Notch and STAT3 inhibitors in cell lines and zebrafish embryos. They observed similar effects on gene expression, cell viability and zebrafish muscle organization. However, this does not mean that triptolide regulates Notch and STAT3 signaling. It is possible that inhibiting other pathways, such as FGF and TGFß, could also produce similar effects. Therefore, major conclusions in this manuscript are not supported by experimental data.

2. The authors did not analyze how triptolide inhibits expression of Notch and STAT3 pathways. Why it decreases the expression of different components of these pathways at the mRNA and protein levels?

3. To claim that triptolide regulates Notch and STAT3 signaling, experiments testing their functional interactions should be performed.

4. The title is mis-leading. It does not reflect the context of the manuscript.

5. There is no statistical significance in figure 2. It is unclear whether there are significant differences between different treatments.

6. In figure 4A, it is not clear whether there are differences between combined and single treatments using Notch and STAT3 pathway inhibitors.

7. The quality of F-actin and Myosin staining is poor. It does not allow to clearly assess the organization of myofibers. It seems that single treatment by P6 causes similar muscle defects as combined treatment (particularly for Myosin staining). It is unclear how the embryos in figure 5C are stained.

8. In contrast to the statement, there is no experiment showing that triptolide interferes with muscle cell proliferation. Disrupted myofiber organization does not mean defects in proliferation. The zebrafish data are not appropriate and are not properly designed to support the claim.

9. Notch signaling is required for somite segmentation. However, treatment with DAPT to inhibit Notch signaling does not cause any somite defect. How do you explain this contradiction?

10. There are no luciferase data for triptolide on Notch and STAT3 signaling.

11. Several descriptions are unclear. In lines 123 and 129, do you mean that Notch1 is a target of Notch1 signaling and STAT3 is a target of STAT3 signaling? The authors should provide references for these statements.

Comments on the Quality of English Language

The manuscript requires proofreading. There are are unclear sentences.

Author Response

Dear reviewer #3

We would like to express our gratitude for your valuable feedback and suggestions. Your comments helped us improve our manuscript significantly in the revision. We have now made every effort to address your concerns more comprehensively in this revised version. We hope that our manuscript accurately reflects the reviewer's points and provides an appropriate response.

We have also attached our responses to other reviewers' comments to help you make decisions about our paper.

Thank you for taking the time to review our work carefully.

Best regards,

Jaekyung Shim, Ph.D.

Assistant professor

Department of Bioresources Engineering, Sejong University,

209, Neungdongro, Kwangjin-Gu, Seoul 05006, Korea

Phone: 82-2-3408-3944 Fax: 82-2-3408-4318

E-mail: jkshim@sejong.ac.kr

Round 2

Reviewer 3 Report (New Reviewer)

Comments and Suggestions for Authors

Major issues raised in the previous round of review were not addressed, and I am not satisfied with most of the responses. In particular, whether TP directly or indirectly affects Notch and STAT3 signaling is not clear. Importantly, it is well established that Notch signaling is involved in somitogenesis. Instead of looking at myofiber organization between 11th and 14th somites, the authors should examine whether perturbation of Notch signaling disrupts somite segmentation.

I am not convinced by the muscle atrophy phenotypes caused by different treatments because the quality of F-actin and Myosin staining is poor and it does not allow to clearly assess the organization of myofibers. Apparently, the authors are unwilling or unable to provide better-quality images.

The authors claim that TP causes muscle defects by inhibiting proliferation. However, they only examined viability of C2C12 cells, whereas whether TP affects muscle cell proliferation was not clear.

This manuscript was submitted to the special issue "Muscle Atrophy”. However, data on muscle analyses are very superficial. Most experiments are performed using cancer cell lines, which are not relevant to the special issue.

Comments on the Quality of English Language

The manuscript requires proofreading. 

Author Response

Dear reviewer

We appreciate your comments and suggestions. In the first revision, we addressed some of the concerns raised by reviewer.

This time, we have tried our best to comprehensively address your feedback to avoid any further questions. We searched for references to solidify our response to the issues raised by the reviewer, particularly the fact that Notch signaling is involved in somitogenesis.

We also replaced the fluorescence image photo with a better one and presented an additional experiment to demonstrate the direct effect of TP on C2C12 cell proliferation.

We hope we have accurately and entirely answered the critical points of this revised manuscript and the reviewers' comments.

Please find attached our second response to the reviewer's comments and our previous first response. Thank you for your help.

Best regards,

Jaekyung Shim, Ph.D.

Department of Bioresources Engineering, Sejong University,

209, Neungdongro, Kwangjin-Gu, Seoul 05006, Korea

Phone: 82-2-3408-3944 Fax: 82-2-3408-4318

E-mail: jkshim@sejong.ac.kr

Round 3

Reviewer 3 Report (New Reviewer)

Comments and Suggestions for Authors

I do not agree with the responses of the authors. In particular, the experimental design does not allow to examine somite segmentation. In zebrafish, the somite segmentation period ends at 22-24 hpf. Of course, you cannot observe segmentation defects (in fact, there is no such an analysis in this work) because somites are already formed when you treat the embryos at 24 hpf.

You mention that you have replaced F-actin and myosin staining with improved images. Specifically for the TP treated condition, it seems that the three images are not from the same embryo.

Comments on the Quality of English Language

Proof editing is necessary.

Author Response

Dear reviewer

We appreciate your third comment on our response and the revised manuscript.

In the first and second revisions, we addressed 16 comments on your issue and presented an appropriate approach to exploring the relationship between muscle loss and TP.

Nonetheless, you raise the issue of ‘somite segmentation' even in your third comment. The attached describes in as much detail as possible the intent of our experiments on this issue.

Additionally, to resolve questions about the fluorescence image photo, the merged photo was replaced with a better photo.

We sincerely hope that this third response will help you make your final decision. If the decrease in the number of comments during the three-round review process reflects the paper's improvement, we would like to request that you revise each checklist related to the paper evaluation.

Thank you once more for the three-round review process.

Best regards,

Jaekyung Shim, Ph.D.

Department of Bioresources Engineering, Sejong University,

209, Neungdongro, Kwangjin-Gu, Seoul 05006, Korea

Phone: 82-2-3408-3944 Fax: 82-2-3408-4318

E-mail: jkshim@sejong.ac.kr

Round 4

Reviewer 3 Report (New Reviewer)

Comments and Suggestions for Authors

The authors are reluctant to improve their manuscript by performing additional experiments. It is well known that triptolide shows global effects on gene expression. Its interaction with Notch and STAT3 signaling in cancers has been extensively studied. There are also many reports that have better characterized the effects of triptolide on muscle cells. Data presented in this manuscript are not novel. Importantly, zebrafish experiments aimed at examining the effects of triptolide on muscle development are not properly designed and the results are not convincing, if not misleading.

The authors replaced images for the TP-treated condition, but they explain that “the quality of the photos may slightly deteriorate in the PDF version of the file”. I find this is dishonest.

Author Response

[The responses for reviewer’s 4th comments]

  1. We responded to the reviewer's concerns by providing logical and appropriate answers based on previous research. Although the reviewer quoted our responses to their questions, they expressed concerns about the lack of novelty in our work.

The introduction of the manuscript aimed to demonstrate the validity and rationale of the study by providing information on the cited research. We specifically highlighted studies that have linked TP with inhibiting the growth of cancer cells and disrupting muscle homeostasis, which may lead to muscle atrophy. In this process, it was confirmed that the anti-proliferative effect of TP can be achieved by suppressing various significant intracellular signals, including at least the Notch1 and STAT3 signaling pathways. We aimed to reveal that this anti-proliferative effect of TP also affects the proliferation of satellite cells (muscle stem cells), which are essential for controlling skeletal muscle life-long maintenance and may cause muscle loss during the final cancer treatment process.

Regarding the same manuscript, the reviewer 1 acknowledges its originality, as shown below, and does not make any comments on it.

“This article possesses slight originality, but would match this special issue.”

2.” Importantly, zebrafish experiments aimed at examining the effects of triptolide on muscle development are not properly designed and the results are not convincing, if not misleading.”

Regarding the reviewer's above opinion, the reviewer failed to acknowledge that muscle development and muscle loss are fundamentally different mechanisms, even though they are interrelated. The reviewer continues to advocate for further experiments on "segmentation," an early muscle development process, despite being advised that it should be addressed separately. This reviewer request appears to be asking for additional experiments beyond what is necessary, possibly due to personal curiosity. However, this experiment is not directly related to muscle loss during adult cancer treatment. “Segmentation” is a major object of muscle development research, as it concerns muscle formation in cases where “somite” continue to exist even after reaching adulthood, such as in insects and fish.

  1. In response to the previous comment about the three images of TP-treated case, we provided precise photographic evidence to counter the comment that the three photographs of the TP-treated sample were not from the same embryo. These images clearly demonstrate that the reviewer's suspicions about the photo were wrong.

However, the viewer disparages the well-intentioned guidance to refer to the original manuscript file if there are any problems during the file conversion process.

This manuscript is a resubmission of an earlier submission. The following is a list of the peer review reports and author responses from that submission.

Round 1

Reviewer 1 Report

Comments and Suggestions for Authors

In the present work, the authors investigated the effects of triptolide (TP) on cancer cell proliferation and muscle atrophy, specifically in the regulation of Notch and STAT3 signaling pathways. Please, see some suggestions and criticisms as follow.

1.     General. Since the aim was to evaluate the effect of TP on muscle atrophy, why the authors evaluate its effect on muscle development (embryos stage) instead muscle mass (adult individuals with already developed skeletal muscle)?

2.     Introduction. The description of previous studies have to be thoroughly improved, detailed, and justified, highlighting the proposition and novelity of the manuscript. Details of the studies cited in the Introduction have to be presented, improving the justification and rationality of the study, specifically the studies about TP and cancer cell proliferation and muscle atrophy. Since there are several previous studies, it is not possible to determine what is new and what is already known about the topic.

3.     Experimental model. Why was mainly used HeLa cells? A brief description of the cell model has to be provided. Please, justify its use, importance, cellular model, biological characteristics, etc.

4.     Methodology. Detail, describe, justify, and give references of the methodology used with Zebrafish. Why was the zebrafish treated at the embryos stage? Why was not in the adult state? It is important to demonstrate the effect in adult stage (instead embryos stage), since the aim is to show the effect in muscle mass, not in muscle development.

5.     Methodology. Justify the sample size. How was it calculated? What is the statistical power of the results? Number of samples, number of experiments, replicates, etc have to be informed.

6.     Methodology. Justify, detail, and give references for all protocols used in the study. How were selected the different cells tested? Why was not evaluated normal cells concomitantly (control cells) with the cancer cells? Justify all concentrations and periods of treatment used.

7.     Results. In the cell viability assay, authors state that TP inhibits cell proliferation, but these processes are independent. What was not evaluated the effect of TP on control/normal cells? This point is important for eliminating deleterious effects of TP. Is TP toxic to the cells (cancer cells or normal cells)?

8.     Additional suggestions. A) Parameters related to inflammation will be interesting to corroborate previous studies. B) Analysis of protein synthesis and degradation pathways will be important to demonstrate the effects of TP on muscle mass in Zebrafish.

9.     Discussion. A) References are lacking in several parts of the Discussion. Please, revise carefully. B) Contradictory effects of TP on muscle atrophy is presented by the authors. How to explain these controversies effects? C) Summarize the risk of bias and limitations of the study. D) Conclusion paragraph has to include the relevance of the work in the research field, perspectives and directions for further studies.

10.  Figure 6 has to be improved. It is not clear what are the final results of the TP treatment. Function of some proteins in the figure is absent.

11.  Minor comments. A) All abbreviations have to be described at the first time that they appear. B) Quality and resolution of the figures have to be improved. Letter and number size is too small. All abbreviations have to be described in the figure legend.

Comments on the Quality of English Language

Minor editing of English language required

Author Response

Dear reviewer1

Thank you very much for taking the time to provide us with your comments and suggestions. We truly appreciate your help in enhancing the revised manuscript.

We have addressed the issues you raised in a point-by-point manner.

After reviewing other reviewers' comments, I have included responses to help you decide on the revised version.

 If you have any further questions or concerns about the revised manuscript, please don't hesitate to contact us.

Best regards,

Jaekyung Shim, Ph.D.

Assistant professor

Department of Bioresources Engineering, Sejong University,

209, Neungdongro, Kwangjin-Gu, Seoul 05006, Korea

Phone: 82-2-3408-3944 Fax: 82-2-3408-4318

E-mail: jkshim@sejong.ac.kr

Reviewer 2 Report

Comments and Suggestions for Authors

Triptolide (TP) is a natural compound in herbal remedies with antiinflammatory and anti-proliferative properties. The authors conducted a study to understand its effects on critical signaling processes within the cell, including Notch1 and STAT3 signaling. Their research showed that TP reduces cancer cell proliferation by decreasing the expression of downstream targets of these signals. Interestingly, inhibiting one signal with a single inhibitor alone did not significantly reduce cancer cell proliferation. They conducted a study on the impact of TP on zebrafish larvae and discovered that it hinders muscle development by suppressing cell proliferation. In addition, they noticed that inhibiting a single type of signaling did not lead to any significant muscle defects. This implies that TP obstructs multiple signals simultaneously, including Notch1 and STAT3, during muscle development. Chemotherapy is commonly used to treat cancer, but it may cause muscle loss due to drugrelated adverse reactions or other complex mechanisms. Their study suggests that anticancer agents like triptolide (TP), inhibiting essential signaling pathways (Notch1 and STAT3), may cause muscle atrophy through antiproliferative activity.

The purpose of this study is clear, and a quick look at the data seems to indicate that there are no problems with the way the data is presented. In fact, there is an inconvenient truth that the author is concealing. First, the Introduction states that Triptolide (TP) may act via STAT3 signaling in cancer cells (original MS, lines 64-65), a fact that has been proven in quite a few papers (see references 1-5). Therefore, there is nothing original in letting TP act on cancer cells and looking at the expression of STAT3. In addition, the fluorescent immunohistological image of zebrafish, which seems to have the greatest originality in this paper, probably did not work well in the experiment itself (Fig. 5A). It should be redone. Furthermore, if you claim that this TP acts through the Notch-1 and STAT3 pathways, you should actually show their expression in zebrafish skeletal muscle by Western blot and fluorescent immunohistochemical staining. Without these data, I strongly suggest that this paper, which confirms only a little originality, should be rejected.

1.     Wang Z et al. (2009) Triptolide downregulates Rac1 and the JAK/STAT3 pathway and inhibits colitis-related colon cancer progression. Mol. Med. 41(10):717-727.

doi: 10.3858/emm.2009.41.10.078.

2.     Huang Y et al. (2019) Triptolide exerts an anti-tumor effect on non‑small cell lung cancer cells by inhibiting activation of the IL‑6/STAT3 axis. J. Mol. Med. 44(1):291-300.

doi: 10.3892/ijmm.2019.4197. 

3.     Zhong Y et al. (2021) Triptolide inhibits JAK2/STAT3 signaling and induces lethal autophagy through ROS generation in cisplatin‑resistant SKOV3/DDP ovarian cancer cells. Oncol Rep

45(5):69.  doi: 10.3892/or.2021.8020. 

4.     Kim JH et al. (2017) Triptolide blocks the STAT3 signaling pathway through induction of protein tyrosine phosphatase SHP-1 in multiple myeloma cells. Int. J. Mol. Med. 40(5):1566-1572.

doi: 10.3892/ijmm.2017.3122. 

5.     Maji S et al. (2019) STAT3- and GSK3β-mediated Mcl-1 regulation modulates TPF resistance in oral squamous cell carcinoma. Carcinogenesis 40(1):173-183. doi: 10.1093/carcin/bgy135.

Author Response

Dear reviewer

Thank you very much for taking the time to provide us with your comments and suggestions. We truly appreciate your help in enhancing the revised manuscript.

We have addressed the issues you raised in a point-by-point manner.

After reviewing other reviewers' comments, I have included responses to help you decide on the revised version.

 If you have any further questions or concerns about the revised manuscript, please don't hesitate to contact us.

Best regards,

Jaekyung Shim, Ph.D.

Assistant professor

Department of Bioresources Engineering, Sejong University,

209, Neungdongro, Kwangjin-Gu, Seoul 05006, Korea

Phone: 82-2-3408-3944 Fax: 82-2-3408-4318

E-mail: jkshim@sejong.ac.kr

Reviewer 3 Report

Comments and Suggestions for Authors

The manuscript is interesting, however, it needs additional improvements before it is eligible for publication.

1. The abstract does not include any text on methods. Also title should state which animal/cell model was used.

2. The introduction is currently written like discussion of the authors previous studies. Please rewrite it. Also the figures with results should not be in the introduction section.

3. Current result section is very long and it contains also high degree of method description and discussion. I suggest to shorten the result section, making it concise and easier to follow. On other hand, the discussion section is too short and merley descriptive. Please make the discussion longer and more thorough.

4. Please state the limiation of your study.

Author Response

(The authors gave the same response as above.)

Round 2

Reviewer 1 Report

Comments and Suggestions for Authors

In the present version of the manuscript, authors have properly addressed most suggestions and criticisms raised in the first revision. However, some questions have to be improved yet.

1.     General. In the manuscript, still it is not clear why the authors evaluate its effect on muscle development (embryos stage) instead muscle mass (adult individuals with already developed skeletal muscle). Since muscle development and muscle atrophy can occur by many different mechanisms, it is not possible to state that these two processes are aways interconnected. The rationality and the proposition of the methodology of the study have to be presented in a clear and direct manner in the “Introduction”, “Methods”, “Discussion” and “Conclusion” sections. The explanation is a little clear in the author’s response letter, but it has to be improved in the manuscript.

2.     Experimental model. Why was mainly used HeLa cells? A brief description of the cell model has to be provided. Please, justify its use, importance, cellular model, biological characteristics, etc. Again, authors presented the rationality of using these cells in the author’s response letter, but any explanation is given in the manuscript.

3.     Methodology. Detail, describe, justify, and give references of the methodology used with Zebrafish. Again, the rationality and proposition of using this model have to be properly presented. References are missing.

4.     Methodology. A general information about the sample size, statistical power of the results, number of samples, number of experiments, replicates, etc have to be informed in the section “Statistical Analysis”.

5.     Methodology. Still missing the justification, details, and references for all protocols used in the study; How the different cells tested were selected; Justification for all concentrations and periods of treatment used.

6.     Additional suggestions. A) Please, in the “Discussion” section, include how the modulation of the inflammatory process observed in previous studies can help to explain the results. B) Please, in the “Discussion” section, include how the modulation of the protein synthesis and degradation pathways observed in previous studies will be important to corroborate the author’s proposition of the effects of TP on muscle mass.

7.    Minor comments. Some figures present imagens and writings overlapped (e.g., Figure 5).

Comments on the Quality of English Language

Minor editing of English language required.

Author Response

Dear reviewers

We would like to express our gratitude for your valuable feedback and suggestions. Your comments helped us improve our manuscript significantly in the first revision. We have now made every effort to address your concerns more comprehensively in this revised version to ensure that there are no further questions. We hope that our manuscript accurately reflects the reviewer's points and provides an appropriate response. Thank you for taking the time to review our work carefully.

Best regards,

Jaekyung Shim, Ph.D.

Assistant professor

Department of Bioresources Engineering, Sejong University,

209, Neungdongro, Kwangjin-Gu, Seoul 05006, Korea

Phone: 82-2-3408-3944 Fax: 82-2-3408-4318

E-mail: jkshim@sejong.ac.kr

Reviewer 2 Report

Comments and Suggestions for Authors

The author does his best to refute this, but it does not change the initial peer review's judgement that this paper has only a small amount of originality. There are also a number of flaws in this paper. An important paper related to this study was not properly cited in the Introduction until we pointed it out. The blurriness of the fluorescent immunohistochemistry of Myosin HC in Fig. 5 is just awful. It is impossible to discern which parts of the tissue are stained. Fluorescent immunohistochemistry of Notch-1 and STAT3, which we requested to ensure the originality of the study, was not carried out because it was difficult to verify. Western blotting of Notch-1, STAT3 or their related substances in skeletal muscle alone, rather than in the whole zebrafish, has also not been carried out. It looks like a smart response by writing a lot of stuff, but it almost doesn't address the claims we have identified as problematic. I therefore judge that this paper does not deserve to be published.

Author Response

Dear reviewers

We apologize that our initial response did not address your concerns adequately. We have taken your second review comment into consideration and have made the necessary efforts to address it. Even if these efforts do not meet your expectations, we will continue to fulfill our scientific obligations by submitting a response to your comment.

Best regards,

Jaekyung Shim, Ph.D.

Assistant professor

Department of Bioresources Engineering, Sejong University,

209, Neungdongro, Kwangjin-Gu, Seoul 05006, Korea

Phone: 82-2-3408-3944 Fax: 82-2-3408-4318

E-mail: jkshim@sejong.ac.kr

Reviewer 3 Report

Comments and Suggestions for Authors

The authors have satisfactorily addressed my comments.

Author Response

Dear reviewers

Thank you for your feedback. Your comments helped improve the manuscript in the first revision. Thank you for taking the time to review our work.

Best regards,

Jaekyung Shim, Ph.D.

Assistant professor

Department of Bioresources Engineering, Sejong University,

209, Neungdongro, Kwangjin-Gu, Seoul 05006, Korea

Phone: 82-2-3408-3944 Fax: 82-2-3408-4318

E-mail: jkshim@sejong.ac.kr

Round 3

Reviewer 1 Report

Comments and Suggestions for Authors

Authors have properly addressed most suggestions and criticisms in the new version of the manuscript. I have no additional main concerns.

Author Response

Thank you for your feedback. Your comments helped improve the manuscript in the revision. Thank you for taking the time to review our work.

Best regards,

Jaekyung Shim, Ph.D.

Assistant professor

Department of Bioresources Engineering, Sejong University,

209, Neungdongro, Kwangjin-Gu, Seoul 05006, Korea

Phone: 82-2-3408-3944 Fax: 82-2-3408-4318

E-mail: jkshim@sejong.ac.kr